# SO(3)-Irreducible Geometry in Complex Dimension Five and Ternary Generalization of Pauli Exclusion Principle

**Viktor Abramov** [1,*] and **Olga Liivapuu** [2]

1   Institute of Mathematics and Statistics, University of Tartu, 51009 Tartu, Estonia
2   Institute of Forestry and Engineering, Estonian University of Life Sciences, 51006 Tartu, Estonia; olga.liivapuu@emu.ee
*   Correspondence: viktor.abramov@ut.ee; Tel.: +372-7375872

**Abstract:** Motivated by a ternary generalization of the Pauli exclusion principle proposed by R. Kerner, we propose a notion of a $\mathbb{Z}_3$-skew-symmetric covariant SO(3)-tensor of the third order, consider it as a 3-dimensional matrix, and study the geometry of the 10-dimensional complex space of these tensors. We split this 10-dimensional space into a direct sum of two 5-dimensional subspaces by means of a primitive third-order root of unity $q$, and in each subspace, there is an irreducible representation of the rotation group $SO(3) \hookrightarrow SU(5)$. We find two SO(3)-invariants of $\mathbb{Z}_3$-skew-symmetric tensors: one is the canonical Hermitian metric in five-dimensional complex vector space and the other is a quadratic form denoted by $K(z, z)$. We study the invariant properties of $K(z, z)$ and find its stabilizer. Making use of these invariant properties, we define an SO(3)-irreducible geometric structure on a five-dimensional complex Hermitian manifold. We study a connection on a five-dimensional complex Hermitian manifold with an SO(3)-irreducible geometric structure and find its curvature and torsion.

**Keywords:** tensors; rotation group; tensor irreducible representations; complex manifolds; special geometries; connection; Pauli exclusion principle

## 1. Introduction

The concept of skew symmetry underlies many structures of modern algebra and geometry. Lie algebras, Grassmann algebras, and algebras of differential forms on smooth manifolds are examples of structures that are based on the concept of skew symmetry. In the case of a Lie algebra, its Lie bracket is skew symmetric with respect to a permutation of arguments of this bracket. A Grassmann algebra can be constructed by means of covariant totally skew-symmetric tensors defined on a finite-dimensional vector space. In this case, the skew symmetry of a tensor means that the rearrangement of any two subscripts leads to a change of the sign of a tensor, and the exterior multiplication of two such tensors is defined with the help of the alternation of the tensor product of these tensors. At the end of the last century and the beginning of this century, we witnessed the emergence of interest towards an *n*-ary generalization of Lie algebra: that is, a generalization in which a Lie bracket contains *n* arguments. The concept of skew symmetry can be easily extended to *n*-ary multiplications if we require that any rearrangement of two arguments in an *n*-ary product results in a change in the sign. An *n*-ary bracket of *n*-Lie algebra is skew symmetric precisely in this classical sense. However, when passing from a binary law of multiplication to an *n*-ary one, when $n > 2$, there arises an interesting question of possible *n*-ary analogues of the classical concept of skew symmetry. In order to formulate the concept of skew symmetry of a multiplication in an algebra, we use permutations of factors in a product. In the case of a binary multiplication, we have only one permutation of variables in a product and, consequently, we have only one notion of a skew-symmetric binary product.

An equivalent form of the condition for skew symmetry of $n$-ary multiplication in the case of a field of real or complex numbers is the requirement that a product of $n$ elements equals zero whenever it contains two equal elements. The notion of skew symmetry given in this form explains why it plays an important role in theoretical physics. It is well known that a fundamental role in quantum physics is played by the Pauli exclusion principle, which states that two fermions in a quantum system cannot co-exist if they have identically equal sets of quantum numbers. It follows then that a wave function of a quantum system containing identical sets of quantum numbers of two fermions must vanish. Now, the skew symmetry of a wave function with respect to the permutation of quantum states of any two fermions follows from the linearity of a wave function.

If we consider an algebra with a ternary multiplication then we have a total of six possible permutations of factors in a ternary product, where three of them are non-cyclic permutations (rearrangements of two factors), and three are cyclic. One way to extend the notion of skew symmetry from binary to ternary multiplication laws is to use non-cyclic permutations: that is, a ternary multiplication is called totally skew-symmetric if it is skew symmetric with respect to any pair of arguments. It is precisely this extension of the notion of skew symmetry that is used in three-Lie algebras, and we will call it the classical ternary skew symmetry. For example, a ternary bracket of a three-Lie algebra changes its sign whenever we rearrange any two elements inside this ternary bracket. Equivalently, if among the three elements of a ternary Lie bracket there are two equal ones, then regardless of where these equal elements appear, a ternary bracket is equal to zero. In this formulation, we see a direct connection with the classical Pauli exclusion principle.

Another way to extend the notion of skew symmetry from binary to ternary multiplication, which we consider in the present paper, is based on cyclic permutations of three elements. The use of cyclic permutations for generalizing the concept of skew symmetry is interesting from the point of view that cyclic permutations form the Abelian group $\mathbb{Z}_3$, while non-cyclic permutations do not. In order to explain how cyclic permutations can be used to extend the concept of skew symmetry to ternary multiplication, consider a general structure: that is, a vector space with a skew-symmetric binary multiplication defined on it. If $x, y$ are elements of this vector space and $x \cdot y$ is the product of these elements, then the skew symmetry can be written in two equivalent ways:

$$x \cdot y + y \cdot x = 0, \tag{1}$$
$$x \cdot y = -y \cdot x. \tag{2}$$

Now assume that we have a complex vector space with ternary multiplication. If $u, v, w$ are elements of this complex vector space, then their ternary product will be denoted by $\tau(u, v, w)$. Since now our aim is to extend the notion of skew symmetry to ternary multiplication by means of cyclic permutations of three elements, it is obvious that a $\mathbb{Z}_3$-analog of (1) is

$$\tau(x, y, z) + \tau(y, z, x) + \tau(z, x, y) = 0. \tag{3}$$

Note that the relation (2) can be interpreted as a solution to the relation (1) in the sense that the product $x \cdot y$ is expressed with the help of $-1$ (which can be considered as the primitive square root of unity) in the terms of the product $y \cdot x$. Analogously, we can solve the relation (3) by making use of the primitive cubic root of unity $q = \exp(2i\pi/3)$. Indeed, if a ternary multiplication $\tau$ satisfies one of the following relations:

$$\tau(x, y, z) = q\, \tau(y, z, x) = \bar{q}\, \tau(z, x, y), \tag{4}$$
$$\tau(x, y, z) = \bar{q}\, \tau(y, z, x) = q\, \tau(z, x, y), \tag{5}$$

then, due to the property $1 + q + \bar{q} = 0$, it also satisfies the relation (3). Thus, each of the relations (4) and (5) can be considered as a $\mathbb{Z}_3$-analog of binary skew symmetry written in the form (2). It is important to note that in the binary case, the relation (1) is equivalent to the relation (2): that is, one immediately follows from the other. This is generally not

true in the ternary case because the relation (3) is not equivalent to the relation (4) or (5). The relation (3) follows from the relation (4) or (5), but not vice-versa. In other words, (3) is a more general condition than (4) or (5).

Thus, we have an approach for extending the concept of skew symmetry from binary to ternary multiplication that differs from the classical one (rearranging two elements changes the sign of a product). This approach uses cyclic permutations of three elements: that is, it is based on the group $\mathbb{Z}_3$ and one of the relations (3)–(5). Here, it should be noted that the approach to ternary skew symmetry based on cyclic permutations differs significantly from the classical approach. Indeed, it is easy to see that no matter which of the conditions (3)–(5) we take as a ternary analogue of the concept of skew symmetry, for any element $x$, we will have $\tau(x, x, x) = 0$. However, the presence of two equal elements in a ternary product, regardless of where they appear in a ternary product, generally does not mean that the whole product will be equal to zero. In this way, $\mathbb{Z}_3$-based ternary skew symmetry differs from classical ternary skew symmetry, wherein a product containing two equal elements vanishes.

An investigation of the above algebraic relations (3)–(5) underlying the $\mathbb{Z}_3$-based ternary concept of skew symmetry was stimulated by a ternary generalization of the Pauli exclusion principle proposed by Richard Kerner [1–4]. A theoretical justification for this ternary generalization of the Pauli exclusion principle is well and thoroughly outlined in the above-mentioned articles by Richard Kerner. Therefore, we will give only a brief description of this principle, and an interested reader will find a detailed description and justification in the articles of Richard Kerner. The ternary generalization of the Pauli exclusion principle proposed by Richard Kerner can be stated as follows: Three particles cannot coexist in a quantum system if these three particles are in the same quantum state, but two such particles can. In the quark model of elementary particles, quarks are considered as fermions, and three quarks or three anti-quarks form a baryon. As an argument in favor of the proposed ternary generalization of the Pauli exclusion principle, Richard Kerner points to the fact that in the quark model, three quarks in the same quantum state cannot form a stable configuration—observed as one of strongly interacting particles—but at the same time, the coexistence of two quarks with the same isospin value is possible. Comparing the ternary generalization of the Pauli exclusion principle formulated in this way with the properties of $\mathbb{Z}_3$-based ternary skew symmetry, we see that $\mathbb{Z}_3$-based ternary skew symmetry is an algebraic structure to give an adequate mathematical description of the ternary generalization of the Pauli exclusion principle.

Relations (3)–(5) can be used to construct a ternary analogue of Grassmann and Clifford algebra: that is, we can consider an algebra over the field of complex numbers generated by a system of generators that obey one of the relations (3)–(5). The properties, structure, and possible applications of such algebras were studied in papers [5–14]. A generalization of the Dirac operator by means of the above-mentioned algebras can be found in [3,4,12,15,16].

In this paper, we study a $\mathbb{Z}_3$-based ternary analog of skew symmetry from a geometric point of view. We consider a complex-valued trilinear form $\tau$ defined on a three-dimensional Euclidean space and take Equation (3) as one of the conditions for the ternary skew symmetry of $\tau$: that is, we assume that $\tau$ satisfies

$$\tau(x, y, z) + \tau(y, z, x) + \tau(z, x, y) = 0. \tag{6}$$

However, if this condition is the only one, then it does not give a very good, so to speak, approximation of classical skew symmetry, for which two equal elements lead to zero. So choosing an orthonormal basis $e_1, e_2, e_3$ for three-dimensional Euclidean space, we impose three more conditions:

$$\sum_{i=1}^{3} \tau(e_i, e_i, x) = 0, \quad \sum_{i=1}^{3} \tau(e_i, x, e_i) = 0, \quad \sum_{i=1}^{3} \tau(x, e_i, e_i) = 0, \tag{7}$$

which are a more general form of the requirement that having two equal elements results in zero. Thus, we say that a trilinear complex-valued form $\tau$ is $\mathbb{Z}_3$-skew-symmetric if it satisfies (6) and (7). It is easy to see that the conditions (6) and (7) of $\mathbb{Z}_3$-skew-symmetry are SO(3)-invariant and do not depend on the choice of an orthonormal basis.

The aim of the present paper is to study a geometry of the space of $\mathbb{Z}_3$-skew-symmetric complex-valued trilinear forms defined on a three-dimensional Euclidean space. The space of $\mathbb{Z}_3$-skew-symmetric complex-valued trilinear forms can be identified with the space of complex-valued third-order covariant SO(3)-tensors that satisfy the conditions

$$T_{ijk} + T_{jki} + T_{kij} = 0, \tag{8}$$

and

$$\sum_{i=1}^{3} T_{iij} = 0, \quad \sum_{i=1}^{3} T_{iji} = 0, \quad \sum_{i=1}^{3} T_{jii} = 0. \tag{9}$$

Let us denote this space by $\mathcal{T}^3$. Then, $\mathcal{T}^3$ is a representation space of a twofold irreducible tensor representation of the rotation group, and the dimension of this space is 10 [17]. A twofold irreducible tensor representation of the rotation group in $\mathcal{T}^3$ splits into two irreducible tensor representations if we decompose the 10-dimensional representation space $\mathcal{T}^3$ into a direct sum of two 5-dimensional subspaces in a way that is invariant under the action of the rotation group. A decomposition into two subspaces can be made with the help of the relations (4) and (5): that is, we define the subspace $\mathcal{T}_q^3 \subset \mathcal{T}^3$ by imposing the additional condition

$$T_{ijk} = q\, T_{jki} = \bar{q}\, T_{kij}. \tag{10}$$

It is easy to see that in this case the condition (8) follows from (10). Hence, we have

$$\mathcal{T}_q^3 = \{T = (T_{ijk}) \in \mathcal{T}^3 : \ T_{ijk} = q\, T_{jki} = \bar{q}\, T_{kij}\}, \tag{11}$$

and analogously,

$$\mathcal{T}_{\bar{q}}^3 = \{T = (T_{ijk}) \in \mathcal{T}^3 : \ T_{ijk} = \bar{q}\, T_{jki} = q\, T_{kij}\}. \tag{12}$$

Then $\mathcal{T}^3 = \mathcal{T}_q^3 \oplus \mathcal{T}_{\bar{q}}^3$, and in each of the subspaces $\mathcal{T}_q^3, \mathcal{T}_{\bar{q}}^3$, we have an irreducible representation of the rotation group. It is known that every representation of the rotation group can be made unitary if we endow a representation space with an appropriate Hermitian metric. We endow the space $\mathcal{T}_q^3$ with the Hermitian metric

$$h(T, S) = T_{ijk}\overline{S}_{ijk} \tag{13}$$

and show that the irreducible representation of the rotation group in $\mathcal{T}_q^3$ is an inclusion $R : \mathrm{SO}(3) \hookrightarrow \mathrm{SU}(5)$. We find the orthonormal basis $E_A, 1 \leq A \leq 5$ (here, $E_A$ are complex-valued third-order covariant tensors satisfying (9) and (10)) for the Hermitian space $\mathcal{T}_q^3$ and identify the space $\mathbb{C}^5$ with the Hermitian vector space of tensors $\mathcal{T}_q^3$ by putting

$$z = (z^A) \in \mathbb{C}^5 \to T(z) = z^A\, E_A \in \mathcal{T}_q^3.$$

Then, the irreducible representation $R$ of the rotation group can be written in the form

$$g_{im}\, g_{jl}\, g_{kp}\, (E_B)_{mlp} = (R(g))_B^A\, (E_A)_{ijk}, \tag{14}$$

where $g = (g_{ij}) \in \mathrm{SO}(3)$. We calculate all SO(3)-invariants of the representation $R$, and this calculation shows that there are only two non-trivial independent invariants. Obviously,

one of them is the canonical Hermitian metric $h(z, \bar{z}) = \sum_A z^A \bar{z}^A$, and the other is the quadratic form

$$K(z, z) = K_{AB} z^A z^B = (z^1)^2 + (z^2)^2 + (z^3)^2 + 2q z^4 z^5. \tag{15}$$

We study the properties of the quadratic form $K(z, z)$. Particularly, we show that the matrix $K_{AB}$ of the quadratic form $K(z, z)$ is symmetric and unitary and that its determinant is the sixth-order primitive root of unity $\epsilon = \exp(\pi i/3)$. These properties are invariant under the action of the unitary group U(5) in the five-dimensional complex space $\mathcal{T}_q^3$. Then, we find the subgroup of the group SU(5) that is a stabilizer of the quadratic form $K(z, z)$ in the five-dimensional complex vector space $\mathcal{T}_q^3$. In analogy to the approach proposed in [18] and developed in [19], we define an SO(3)-irreducible geometric structure in complex dimension five and study its geometry.

## 2. Five-Dimensional Complex Space of the SO(3)-Irreducible Representation

The aim of this section is to describe an irreducible tensor representation of the rotation group. In what follows, we consider complex-valued covariant tensors defined in three-dimensional Euclidean space $\mathbb{R}^3$. Let $T = (T_{i_1 i_2 \ldots i_p})$ be a tensor of rank $p$. In what follows, we will use the Einstein convention of summation over repeated indices. Then, the formula

$$\tilde{T}_{j_1 j_2 \ldots j_p} = g_{i_1 j_1} g_{i_2 j_2} \cdots g_{i_p j_p} T_{i_1 i_2 \ldots i_p}, \tag{16}$$

where $g = (g_{ij}) \in \mathrm{SO}(3)$ is a rotation in $\mathbb{R}^3$, defines a linear transformation in a vector space of covariant tensors of rank $p$, i.e., it defines a representation of the rotation group, which is called a 'tensor representation'. A linear transformation (16) will be denoted by $g \cdot T$: that is, $\tilde{T} = g \cdot T$. In this section, we give an explicit description of an irreducible five-dimensional tensor representation of the rotation group in the complex vector space of covariant tensors of rank three.

Let $\mathcal{T}^3$ be the vector space of tensors of rank three that satisfy the following conditions:

T1. A contraction of a tensor $T = (T_{ijk})$ over any pair of subscripts (trace) is zero: that is, for any $j = 1, 2, 3$, it holds that

$$T_{iij} = 0, \quad T_{iji} = 0, \quad T_{jii} = 0.$$

T2. For any combination of integers $i, j, k$ (each running from 1 to 3), the sum of the components of tensor $T = (T_{ijk})$ obtained by cyclic permutation of its subscripts is equal to zero: that is,

$$T_{ijk} + T_{jki} + T_{kij} = 0.. \tag{17}$$

It can be easily verified that the conditions $\mathrm{T}_1, \mathrm{T}_2$ are invariant under the action of the rotation group (16). Hence, for any rotation $g \in \mathrm{SO}(3)$, we have $R_g : \mathcal{T}^3 \to \mathcal{T}^3$. It is shown in [17] that the vector space $\mathcal{T}^3$ is 10-dimensional, and the Formula (16) defines a twofold irreducible tensor representation of the rotation group in this vector space. If we split the 10-dimensional vector space $\mathcal{T}^3$ into a direct sum of two 5-dimensional subspaces in a way so that they are invariant with respect to the action of the rotation group (16); then, in each 5-dimensional subspace of $\mathcal{T}^3$, we will have an irreducible tensor representation of the rotation group.

One can split the 10-dimensional vector space $\mathcal{T}^3$ into a direct sum of two 5-dimensional subspaces that are invariant with respect to a tensor representation of the rotation group by making use of a linear operator induced by a substitution. Let us denote by $\sigma$ the cyclic substitution of the first three integers $\sigma(1) = 2, \sigma(2) = 3, \sigma(3) = 1$. Then, one can define the operator $\Phi_\sigma : T \to \tilde{T}$ acting on the tensors of rank three as follows:

$$\tilde{T}_{i_1 i_2 i_3} = \Phi_\sigma(T_{i_1 i_2 i_3}) = T_{i_{\sigma(1)} i_{\sigma(2)} i_{\sigma(3)}} = T_{i_2 i_3 i_1},$$

and extend this by linearity to the vector space of all tensors of rank three. Obviously, $\Phi_\sigma^3 = \mathrm{id}$, where id is the identity transformation, and

$$(\mathrm{id} + \Phi_\sigma + \Phi_\sigma^2)(T_{ijk}) = T_{ijk} + T_{jki} + T_{kij}, \qquad (18)$$

Thus, Equation (17) can be written in the form

$$(\mathrm{id} + \Phi_\sigma + \Phi_\sigma^2)(T) = 0. \qquad (19)$$

Now, it is easy to show that the vector space $\mathcal{T}^3$ is invariant under the action of the operator $\Phi_\sigma$: that is, $\Phi_\sigma : \mathcal{T}^3 \to \mathcal{T}^3$. Assume that a tensor $T = (T_{ijk})$ satisfies the condition $\mathrm{T}_2$ or, equivalently, Equation (19). Denote $\tilde{T} = \Phi_\sigma(T)$. Then,

$$(\mathrm{id} + \Phi_\sigma + \Phi_\sigma^2)(\tilde{T}) = (\mathrm{id} + \Phi_\sigma + \Phi_\sigma^2)(\Phi_\sigma(T)) = (\mathrm{id} + \Phi_\sigma + \Phi_\sigma^2)(T) = 0,$$

and $\tilde{T}$ also satisfies Equation (19). Similarly, one can verify that the operator $\Phi_\sigma$ preserves the condition $\mathrm{T}_1$.

Generally, the property of the linear operator $\Phi_\sigma^3 = \mathrm{id}$ implies that it has three eigenvalues $1, q, \bar{q}$ in the vector space of all tensors of rank three. Here, $q = \exp(2i\pi/3)$ is the primitive third-order root of unity and $\bar{q}$ is its complex conjugate. Another general formula is based on the property of the third-order roots of unity $1 + q + \bar{q} = 0$. Indeed, it is easy to see that due to the mentioned property of the third-order roots of unity, any tensor of rank three can be decomposed into the sum of three tensors:

$$T = T_1 + T_q + T_{\bar{q}}, \qquad (20)$$

where

$$
\begin{aligned}
T_1 &= \frac{1}{3}(\mathrm{id} + \Phi_\sigma + \Phi_\sigma^2)(T) && \text{or} \quad (T_1)_{ijk} = \frac{1}{3}(T_{ijk} + T_{jki} + T_{kij}), \\
T_q &= \frac{1}{3}(\mathrm{id} + \bar{q}\,\Phi_\sigma + q\,\Phi_\sigma^2)(T) && \text{or} \quad (T_q)_{ijk} = \frac{1}{3}(T_{ijk} + \bar{q}\,T_{jki} + q\,T_{kij}), \\
T_{\bar{q}} &= \frac{1}{3}(\mathrm{id} + q\,\Phi_\sigma + \bar{q}\,\Phi_\sigma^2)(T) && \text{or} \quad (T_{\bar{q}})_{ijk} = \frac{1}{3}(T_{ijk} + q\,T_{jki} + \bar{q}\,T_{kij}).
\end{aligned}
$$

Obviously, the tensors $T_1, T_q, T_{\bar{q}}$ are the eigenvectors of the linear operator $\Phi_\sigma$ corresponding to the eigenvalues $1, q, \bar{q}$, respectively. Thus, we have

$$\Phi_\sigma(T_1) = T_1, \quad \Phi_\sigma(T_q) = q\,T_q, \quad \Phi_\sigma(T_{\bar{q}}) = \bar{q}\,T_{\bar{q}},$$

or, equivalently,

$$(T_1)_{ijk} = (T_1)_{jki}, \quad (T_q)_{ijk} = \bar{q}\,(T_q)_{jki}, \quad (T_{\bar{q}})_{ijk} = q\,(T_{\bar{q}})_{jki}.$$

It is worth mentioning that the components $T_q$ and $T_{\bar{q}}$ of a tensor $T$ satisfy the condition $\mathrm{T}_2$. Restricting (20) to the vector space $\mathcal{T}^3$, we see that due to the condition $\mathrm{T}_2$, the first term on the right-hand side vanishes, i.e., $T_1 = 0$, and (20) takes on the form $T = T_q + T_{\bar{q}}$, where $T_q, T_{\bar{q}} \in \mathcal{T}^3$. Hence, we can decompose the vector space $\mathcal{T}^3$ into the direct sum of two subspaces, which are denoted as $\mathcal{T}_q^3$ and $\mathcal{T}_{\bar{q}}^3$. According to the definitions (11) and (12), $\mathcal{T}_q^3$ is the subspace of the eigenvectors of the linear operator $\Phi_\sigma$ with eigenvalue $\bar{q}$, and $\mathcal{T}_{\bar{q}}^3$ is the subspace of the eigenvectors of the linear operator $\Phi_\sigma$ with eigenvalue $q$. Thus, $\mathcal{T}^3 = \mathcal{T}_q^3 \oplus \mathcal{T}_{\bar{q}}^3$.

The subspaces $\mathcal{T}_q^3, \mathcal{T}_{\bar{q}}^3$ play a basic role in what follows, and it is useful to give their exact description here. $\mathcal{T}_q^3$ is a vector space of complex-valued tensors of rank three that satisfy the condition $\mathrm{T}_1$ (a trace over any pair of subscripts is zero), and the tensors are eigenvectors of the linear operator $\Phi_\sigma$ with eigenvalue $\bar{q}$: that is, they satisfy $\Phi_\sigma(T) = \bar{q}\,T$

or $T_{ijk} = q\, T_{jki}$. Similarly, $\mathcal{T}^3_{\bar{q}}$ is a vector space of complex-valued tensors of rank three that satisfy $T_1$, and the tensors are the eigenvectors of the linear operator $\Phi_\sigma$ with eigenvalue $q$, i.e., $\Phi_\sigma(T) = q\, T$ or $T_{ijk} = \bar{q}\, T_{jki}$. Hence,

$$\mathcal{T}^3_q = \{T \in \mathcal{T}^3 : \Phi_\sigma(T) = \bar{q}\, T\}, \ \ \mathcal{T}^3_{\bar{q}} = \{T \in \mathcal{T}^3 : \Phi_\sigma(T) = q\, T\}. \tag{21}$$

The important role of these subspaces is that they are spaces of a five-dimensional irreducible representation of the rotation group.

A tensor of the third rank $T = (T_{ijk})$ is a quantity with three subscripts $i, j, k$. Therefore, in what follows, it will be convenient for us to represent tensors of the third rank in the form of three-dimensional matrices, which are also called hypermatrices. By a three-dimensional matrix, we mean a three-dimensional cube with the components of a tensor $T_{ijk}$ located on the sections of this cube. Here, by section, we mean a section of a cube by a plane perpendicular to its edges. We assume that the cube is located in space so that the first subscript $i$ of a tensor $T_{ijk}$ enumerates sections of a cube parallel to the plane of this page, and the numbering starts from the section closest to us ($i = 1$) and then takes values $2, 3$ as the distance from us increases (see Figure 1).

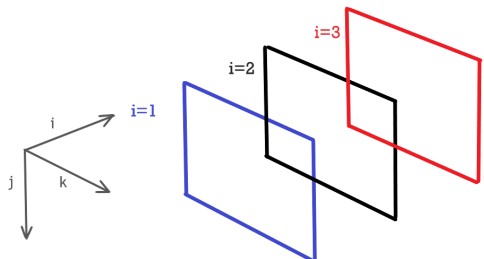

**Figure 1.** Tensor components arranged as a three-dimensional matrix.

We map a three-dimensional matrix onto the plane of the page of this paper by placing the numbered sections of a cube (which are the usual third-order square matrices) from left to right: that is, on the left, there is the section with $i = 1$, the center is $i = 2$, and the right is $i = 3$. Thus, a three-dimensional matrix of a third-order tensor $T = (T_{ijk})$ can be represented as follows:

$$T = \left( \begin{array}{ccc|ccc|ccc} T_{111} & T_{112} & T_{113} & T_{211} & T_{212} & T_{213} & T_{311} & T_{312} & T_{313} \\ T_{121} & T_{122} & T_{123} & T_{221} & T_{222} & T_{223} & T_{321} & T_{322} & T_{323} \\ T_{131} & T_{132} & T_{133} & T_{231} & T_{232} & T_{233} & T_{331} & T_{332} & T_{333} \end{array} \right). \tag{22}$$

If a three-dimensional matrix $T$ is represented in the form (22), then we say that $T = (T_{ijk})$ is written in the direction $i$. Analogously, we can define $j$-directional and $k$-directional representations of a three-dimensional matrix.

Now, we consider the five-dimensional complex vector space $\mathbb{C}^5$ endowed with the canonical Hermitian metric $h$. The coordinates of this space are denoted by $z^A$, where $A$ runs from 1 to 5. Then, $h(z, \bar{z}) = \sum_{A=1}^{5} z^A \bar{z}^A$. We identify this five-dimensional complex vector space with the complex vector space of the third-rank covariant tensors (or three-dimensional matrices) $\mathcal{T}^3_q$ by identifying a point $(z^A) \in \mathbb{C}^5$ with the three-dimensional matrix $T(z)$, i.e.,

$$(z^A) \in \mathbb{C}^5 \ \rightarrow \ T(z) = (T_{ijk}(z)) \in \mathcal{T}^3_q,$$

where

$$T(z) = \begin{pmatrix} 0 & -\frac{\bar{q}z^2}{\sqrt{6}} & \frac{\bar{q}z^3}{\sqrt{6}} & -\frac{z^2}{\sqrt{6}} & \frac{q z^1}{\sqrt{6}} & \frac{\bar{q}z^5}{\sqrt{3}} & \frac{z^3}{\sqrt{6}} & \frac{q z^4}{\sqrt{3}} & -\frac{q z^1}{\sqrt{6}} \\ -\frac{q z^2}{\sqrt{6}} & \frac{z^1}{\sqrt{6}} & \frac{z^4}{\sqrt{3}} & \frac{\bar{q}z^1}{\sqrt{6}} & 0 & -\frac{\bar{q}z^3}{\sqrt{6}} & \frac{z^5}{\sqrt{3}} & -\frac{z^3}{\sqrt{6}} & \frac{q z^2}{\sqrt{6}} \\ \frac{q z^3}{\sqrt{6}} & \frac{q z^5}{\sqrt{3}} & -\frac{z^1}{\sqrt{6}} & \frac{\bar{q}z^4}{\sqrt{3}} & -\frac{q z^3}{\sqrt{6}} & \frac{z^2}{\sqrt{6}} & -\frac{\bar{q}z^1}{\sqrt{6}} & \frac{\bar{q}z^2}{\sqrt{6}} & 0 \end{pmatrix}. \quad (23)$$

It is easy to verify that $T(z)$ satisfies the condition $T_1$: that is, the trace over any pair of subscripts is zero, and $T_{ijk}(z) = q\, T_{jki}(z)$. Thus, in what follows, we consider the five-dimensional complex vector space for which the points are identified with three-dimensional complex matrices $T(z)$.

The Formula (16) defines an action $T \to g \cdot T$ of the rotation group SO(3) on our five-dimensional complex vector space, and this action yields an irreducible tensor representation of the rotation group [17]. Now our aim is to find and study the invariants of this representation, which we will use to construct an irreducible special geometry.

In this paper, we use the classification of invariants of third-order tensors that transform according to Formula (16): that is, under the action of the rotation group SO(3). This classification can be found in [20]. If we do not assume that a tensor $T = (T_{ijk})$ has any symmetries, then there is only one linear invariant:

$$I = \epsilon_{ijk}\, T_{ijk} = T_{123} + T_{231} + T_{312} - T_{321} - T_{213} - T_{132},$$

where $\epsilon_{ijk}$ is the Levi–Civita tensor in three-dimensional Euclidean space. Since we consider the space of tensors that satisfy property $T_2$—that is, the sum of the components obtained by cyclic permutations of subscripts is equal to zero—the linear invariant $I_1$ vanishes.

The complete set of quadratic SO(3)-invariants of a third-order real-valued tensor $T$ (no symmetries) includes eleven invariants. Six of the eleven invariants contain the trace of a tensor $T$ with respect to some pair of subscripts, and, due to property $T_1$, these invariants vanish when restricted to the space $\mathcal{T}_q^3$. This leaves us with five SO(3)-invariants of a real-valued tensor, and these invariants are on the left side of the table shown below. Due to the fact that we are considering complex-valued tensors, this list of five invariants should be extended by supplementing it with additional invariants. These additional invariants are constructed from those on the left side of the table by replacing one of the factors in a product of tensor components with the complex conjugate, and the additional invariants are shown on the right side of the table. Direct calculation leads to the following table of invariants:

$$I_1 = T_{ijk}\, T_{ijk} = 0, \qquad\qquad I_1^* = T_{ijk}\, \overline{T}_{ijk} = \sum_{A=1}^{5} z^A\, \bar{z}^A = h(z, \bar{z}), \quad (24)$$

$$I_2 = T_{ijk}\, T_{ikj} = \sum_{A=1}^{3} (z^A)^2 + 2\, q\, z^4 z^5, \quad I_2^* = T_{ijk}\, \overline{T}_{ikj} = 0, \quad\quad\qquad (25)$$

$$I_3 = T_{ijk}\, T_{jik} = q\, T_{ijk}\, T_{ikj} = q\, I_2, \qquad I_3^* = T_{ijk}\, \overline{T}_{jik} = \bar{q}\, T_{ijk}\, \overline{T}_{ikj} = \bar{q}\, I_2^* = 0, \quad (26)$$

$$I_4 = T_{ijk}\, T_{kji} = \bar{q}\, T_{ijk} T_{ikj} = \bar{q}\, I_2, \qquad I_4^* = T_{ijk}\overline{T}_{kji} = q\, T_{ijk}\overline{T}_{ikj} = q\, I_2^* = 0, \quad (27)$$

$$I_5 = T_{ijk} T_{kij} + T_{ijk} T_{jki} = 0, \qquad I_5^* = T_{ijk}\overline{T}_{kij} + T_{ijk}\overline{T}_{jki} = -I_1^*. \quad (28)$$

The table of invariants shows that we have two independent quadratic invariants $I_1^*, I_2$, with the first one $I_1^*$ being the canonical Hermitian metric of the complex five-dimensional space $\mathbb{C}^5$. Hence, each rotation of the three-dimensional space $\mathbb{R}^3$ induces a unitary transformation of the complex five-dimensional Hermitian space $\mathbb{C}^5$: that is, we have a representation $R : g \in$ SO(3) $\to R_g \in$ U(5). Obviously the homomorphism $R$ from the rotation group into the group of unitary matrices of order five is injective. At the infinitesimal level, the representation $R$ generates the representation of the Lie algebra of the rotation group $\varrho : \tau \in \mathfrak{so}(3) \to \varrho_\tau \in \mathfrak{u}(5)$. Our next goal is to find an explicit form

of this representation using the basis of the five-dimensional complex Hermitian space of three-dimensional matrices (23). In other words, given a skew-symmetric third-order matrix $\tau = (\tau_{ij}) \in \mathfrak{so}(3)$, we will calculate a fifth-order skew-Hermitian traceless matrix $\varrho_\tau = ((\varrho_\tau)^A_B) \in \mathfrak{u}(5)$. We see that the form of the skew-Hermitian matrix $\varrho_\tau$ is determined by the second invariant $I_2$, and this matrix is surprisingly similar to the matrix used in the Georgi–Glashow model for unification of elementary particles [21].

In order to calculate the infinitesimal part of the representation $R : \mathrm{SO}(3) \to \mathrm{U}(5)$, we use the exponential map from the Lie algebra $\mathfrak{so}(3)$ to the rotation group $\mathrm{SO}(3)$; we take only the linear part of the corresponding expansion $g_{ij} = \delta_{ij} + \tau_{ij} + \ldots$, where $\tau = (\tau_{ij})$ is a skew-symmetric matrix. We can write

$$
\begin{aligned}
(g \cdot T)_{ijk} &= g_{ip}g_{jr}g_{ks}T_{prs} = (\delta_{ip} + \tau_{ip} + \ldots)(\delta_{jr} + \tau_{jr} + \ldots)(\delta_{ks} + \tau_{ks} + \ldots)T_{prs} \\
&= T_{ijk} + \tau_{ip}T_{pjk} + \tau_{jr}T_{irk} + \tau_{ks}T_{ijs} + \ldots
\end{aligned}
\tag{29}
$$

Hence, the infinitesimal part of the action $T \to g \cdot T$ (up to the terms of the second order and higher) defines the linear operator $\varrho_\tau : \mathbb{C}^5 \to \mathbb{C}^5$, where

$$
(\varrho_\tau(T))_{ijk} = \tau_{ip}T_{pjk} + \tau_{jr}T_{irk} + \tau_{ks}T_{ijs}.
\tag{30}
$$

It will be convenient for us to pass to a parameterization of matrix $\tau = (\tau_{ij})$ with the help of parameters containing one index. Let us define $\tau_i = -\frac{1}{2}\epsilon_{ijk}\tau_{jk}$. Then

$$
\tau = \begin{pmatrix} 0 & -\tau_3 & \tau_2 \\ \tau_3 & 0 & -\tau_1 \\ -\tau_2 & \tau_1 & 0 \end{pmatrix}
$$

Now we can calculate a matrix of this operator (we will use the same notation $\varrho_\tau$ for the matrix of the operator) by means of the following basis in five-dimensional complex space of three-dimensional matrices (23):

$$
\left( \begin{array}{ccc|ccc|ccc} 0 & 0 & 0 & 0 & q & 0 & 0 & 0 & -q \\ 0 & 1 & 0 & \bar{q} & 0 & 0 & 0 & 0 & 0 \\ 0 & 0 & -1 & 0 & 0 & 0 & -\bar{q} & 0 & 0 \end{array} \right),
\left( \begin{array}{ccc|ccc|ccc} 0 & -\bar{q} & 0 & -1 & 0 & 0 & 0 & 0 & 0 \\ -q & 0 & 0 & 0 & 0 & 0 & 0 & 0 & q \\ 0 & 0 & 0 & 0 & 0 & 1 & 0 & \bar{q} & 0 \end{array} \right),
$$

$$
\left( \begin{array}{ccc|ccc|ccc} 0 & 0 & \bar{q} & 0 & 0 & 0 & 1 & 0 & 0 \\ 0 & 0 & 0 & 0 & 0 & -\bar{q} & 0 & -1 & 0 \\ q & 0 & 0 & 0 & -q & 0 & 0 & 0 & 0 \end{array} \right),
\left( \begin{array}{ccc|ccc|ccc} 0 & 0 & 0 & 0 & 0 & 0 & 0 & \sqrt{2}q & 0 \\ 0 & 0 & \sqrt{2} & 0 & 0 & 0 & 0 & 0 & 0 \\ 0 & 0 & 0 & \sqrt{2}\bar{q} & 0 & 0 & 0 & 0 & 0 \end{array} \right),
$$

$$
\left( \begin{array}{ccc|ccc|ccc} 0 & 0 & 0 & 0 & 0 & \sqrt{2}\bar{q} & 0 & 0 & 0 \\ 0 & 0 & 0 & 0 & 0 & 0 & \sqrt{2} & 0 & 0 \\ 0 & \sqrt{2}q & 0 & 0 & 0 & 0 & 0 & 0 & 0 \end{array} \right),
$$

Let us denote the three-dimensional matrices of this basis by $E_A$, where $A = 1, 2, \ldots, 5$. By straightforward calculation, we find

$$
\begin{aligned}
\varrho_\tau(E_1) &= -\tau_3\, E_2 + \tau_2\, E_3 + \sqrt{2}\tau_1\, E_4 + \sqrt{2}\,\bar{q}\,\tau_1\, E_5, \\
\varrho_\tau(E_2) &= \tau_3\, E_1 - \tau_1\, E_3 + \sqrt{2}\,q\,\tau_2\, E_4 + \sqrt{2}\,q\,\tau_2\, E_5, \\
\varrho_\tau(E_3) &= -\tau_2\, E_1 + \tau_1\, E_2 + \sqrt{2}\,\bar{q}\,\tau_3\, E_4 + \sqrt{2}\,\tau_3\, E_5, \\
\varrho_\tau(E_4) &= -\sqrt{2}\,\tau_1\, E_1 - \sqrt{2}\,\bar{q}\,\tau_2\, E_2 - \sqrt{2}\,q\,\tau_3\, E_3, \\
\varrho_\tau(E_5) &= -\sqrt{2}\,q\,\tau_1\, E_1 - \sqrt{2}\,\bar{q}\,\tau_2\, E_2 - \sqrt{2}\,\tau_3\, E_3.
\end{aligned}
$$

Hence, the matrix of the operator $\varrho_\tau$ has the form

$$\varrho_\tau = \begin{pmatrix} 0 & \tau_3 & -\tau_2 & -\sqrt{2}\,\tau_1 & -\sqrt{2}\,q\,\tau_1 \\ -\tau_3 & 0 & \tau_1 & -\sqrt{2}\,\bar{q}\,\tau_2 & -\sqrt{2}\,\bar{q}\,\tau_2 \\ \tau_2 & -\tau_1 & 0 & -\sqrt{2}\,q\,\tau_3 & -\sqrt{2}\,\tau_3 \\ \sqrt{2}\,\tau_1 & \sqrt{2}\,q\,\tau_2 & \sqrt{2}\,\bar{q}\,\tau_3 & 0 & 0 \\ \sqrt{2}\,\bar{q}\,\tau_1 & \sqrt{2}\,q\,\tau_2 & \sqrt{2}\,\tau_3 & 0 & 0 \end{pmatrix}. \tag{31}$$

Due to the fact that the irreducible representation of the rotation group in the complex space of three-dimensional matrices (23) is unitary (as we mentioned above, one of the invariants of this representation is the Hermitian metric of the five-dimensional complex space), the matrix of the representation of the Lie algebra of the rotation group $\varrho_\tau$ must be skew-Hermitian, and this is indeed the case, because the matrix $\varrho_\tau$ satisfies the relation $\varrho_\tau + \varrho_\tau^\dagger = 0$, where $\varrho_\tau^\dagger = \bar{\varrho}_\tau^{\mathrm{T}}$. It is easy to see that $\rho_\tau$ is a traceless matrix. Hence, $\rho_\tau$ belongs to the Lie algebra of the group SU(5): that is, $\rho_\tau \in \mathfrak{su}(5)$. Hence, we can express this matrix in terms of generators of $\mathfrak{su}(5)$, which are denoted in physics papers by $L_i$, where $i = 1, 2, \ldots, 24$, and $L_i$ are Hermitian traceless matrices of the fifth-order normalized by $\mathrm{Tr}(L_i L_j) = \frac{1}{2}\delta_{ij}$. In this paper, we use the following numbering for the generators of $\mathfrak{su}(5)$:

- The first eight generators correspond to SU(3): that is,

$$L_k = \frac{1}{2}\begin{pmatrix} \lambda_k & 0 \\ 0 & 0 \end{pmatrix},$$

  where $\lambda_i$ are Gell–Mann matrices;
- the next four generators $L_9, L_{10}, L_{11}, L_{12}$ have the form

$$L_{8+k} = \frac{1}{2}\begin{pmatrix} 0 & 0 \\ 0 & \sigma_k \end{pmatrix}, \quad L_{12} = \frac{1}{2\sqrt{15}}\,\mathrm{Diag}(-2,-2,-2,3,3),$$

  where $k = 1, 2, 3$ and $\sigma_1, \sigma_2, \sigma_3$ are Pauli matrices;
- the next twelve generators (sometimes called broken matrices) are of the form

$$\begin{aligned} L_{12+k} &= \frac{1}{2}(d_4^k + d_k^4), & L_{15+k} &= \frac{1}{2}(d_5^k + d_k^5), \\ L_{18+k} &= -\frac{i}{2}(d_4^k - d_k^4), & L_{21+k} &= -\frac{i}{2}(d_5^k - d_k^5), \end{aligned}$$

  where $k = 1, 2, 3$, and $d_k^i$ is a matrix with only one non-zero element, which is at the intersection of $i$th row with the $k$th column.

Then, the matrix $\rho_\tau$ can be written in the terms of $\mathfrak{su}(5)$-generators $L_i$ as follows:

$$\begin{aligned} \rho_\tau =\ & 2i\,\tau_1\left(L_7 - \frac{\sqrt{6}}{2}L_{16} - \sqrt{2}\,L_{19} + \frac{\sqrt{2}}{2}L_{22}\right) \\ &+ 2i\,\tau_2\left(-L_5 + \frac{\sqrt{6}}{2}L_{14} - \frac{\sqrt{6}}{2}L_{17} + \frac{\sqrt{2}}{2}L_{20} + \frac{\sqrt{2}}{2}L_{23}\right) \\ &+ 2i\,\tau_3\left(L_2 - \frac{\sqrt{6}}{2}L_{15} + \frac{\sqrt{2}}{2}L_{21} - \sqrt{2}\,L_{24}\right). \end{aligned}$$

It should be noted here that the matrix $\rho_\tau$ is not only skew-Hermitian and traceless, it also satisfies some additional conditions that follow from the fact that the irreducible representation of the rotation group has one more quadratic invariant $I_2$ (25). We denote the quadratic form in the five-dimensional complex vector space induced by this invariant as follows:

$$K(z,z) = K_{AB}\,z^A z^B = (z^1)^2 + (z^2)^2 + (z^3)^{2\prime} + 2\,q\,z^4 z^5. \tag{32}$$

The matrix of this quadratic form

$$K = (K_{AB}) = \begin{pmatrix} 1 & 0 & 0 & 0 & 0 \\ 0 & 1 & 0 & 0 & 0 \\ 0 & 0 & 1 & 0 & 0 \\ 0 & 0 & 0 & 0 & q \\ 0 & 0 & 0 & q & 0 \end{pmatrix}, \tag{33}$$

can be considered as a covariant second-order tensor in the five-dimensional complex vector space, and the properties of this tensor will be studied in the next section. Here, we only note that the matrix $K = (K_{AB})$ is symmetric and unitary, i.e.,

$$K = K^{\mathrm{T}}, \quad K K^{\dagger} = E, \tag{34}$$

where $E = (\delta_{AB})$ is the identity matrix.

The infinitesimal action (30) generates the following vector fields in five-dimensional complex space:

$$\begin{aligned}
X_1 &= (\sqrt{2}\,z^4 + \sqrt{2}\,q\,z^5)\frac{\partial}{\partial z^1} - z^3\frac{\partial}{\partial z^2} + z^2\frac{\partial}{\partial z^3} - \sqrt{2}\,z^1\frac{\partial}{\partial z^4} - \sqrt{2}\,\bar{q}\,z^1\frac{\partial}{\partial z^5}, \\
X_2 &= z^3\frac{\partial}{\partial z^1} + \sqrt{2}\,\bar{q}\,(z^4 + z^5)\frac{\partial}{\partial z^2} - z^1\frac{\partial}{\partial z^3} - \sqrt{2}\,q\,z^2\frac{\partial}{\partial z^4} - \sqrt{2}\,q\,z^2\frac{\partial}{\partial z^5}, \\
X_2 &= -z^2\frac{\partial}{\partial z^1} + z^1\frac{\partial}{\partial z^2} + \sqrt{2}(q\,z^4 + z^5)\frac{\partial}{\partial z^3} - \sqrt{2}\,\bar{q}\,z^3\frac{\partial}{\partial z^4} - \sqrt{2}\,z^3\frac{\partial}{\partial z^5}.
\end{aligned}$$

These vector fields span the Lie algebra $[X_1, X_2] = X_3, [X_2, X_3] = X_1, [X_3, X_1] = X_2$ isomorphic to the Lie algebra of matrices (31). Due to the fact that the Hermitian metric $h(z, \bar{z})$ and the quadratic form $K(z, z)$ are invariants of the irreducible representation of the rotation group $g \in \mathrm{SO}(3) \to R_g \in \mathrm{U}(5)$, the vector fields $X_1, X_2, X_3$ vanish on the Hermitian form $h(z, \bar{z})$ and the quadratic form $K(z, z)$.

Now our goal is to show that, in fact, the irreducible representation of the rotation group $g \in \mathrm{SO}(3) \to R_g \in \mathrm{U}(5)$ has the form $g \in \mathrm{SO}(3) \to R_g \in \mathrm{SU}(5)$: that is, each rotation generates a unitary with determinant 1 transformation in the five-dimensional complex vector space. For this purpose, we will find a parameterization of the irreducible representation using Euler angles. Let us consider one-parameter subgroups of the rotation group

$$g_1(t) = \begin{pmatrix} \cos t & -\sin t & 0 \\ \sin t & \cos t & 0 \\ 0 & 0 & 1 \end{pmatrix}, \quad g_2(t) = \begin{pmatrix} 1 & 0 & 0 \\ 0 & \cos t & -\sin t \\ 0 & \sin t & \cos t \end{pmatrix}. \tag{35}$$

The one-parameter subgroups of unitary transformations in five-dimensional complex vector space generated by the irreducible representation of $g_1(t)$ and $g_2(t)$ have the following forms, respectively:

$$R_1(t) = \begin{pmatrix} \cos t & \sin t & 0 & 0 & 0 \\ -\sin t & \cos t & 0 & 0 & 0 \\ 0 & 0 & \cos 2t & -\frac{q}{\sqrt{2}}\sin 2t & -\frac{1}{\sqrt{2}}\sin 2t \\ 0 & 0 & \frac{\bar{q}}{\sqrt{2}}\sin 2t & \cos^2 t & -\bar{q}\sin^2 t \\ 0 & 0 & \frac{1}{\sqrt{2}}\sin 2t & -q\sin^2 t & \cos^2 t \end{pmatrix}, \tag{36}$$

$$R_2(t) = \begin{pmatrix} \cos 2t & 0 & 0 & -\frac{1}{\sqrt{2}}\sin 2t & -\frac{q}{\sqrt{2}}\sin 2t \\ 0 & \cos t & \sin t & 0 & 0 \\ 0 & -\sin t & \cos t & 0 & 0 \\ \frac{1}{\sqrt{2}}\sin 2t & 0 & 0 & \cos^2 t & -q\sin^2 t \\ \frac{\bar{q}}{\sqrt{2}}\sin 2t & 0 & 0 & -\bar{q}\sin^2 t & \cos^2 t \end{pmatrix}. \tag{37}$$

Direct calculation shows that the determinants of these matrices are equal to 1. Since any rotation can be written as a composition $g_1(\phi)\, g_2(\theta)\, g_1(\psi)$, where $\phi, \theta, \psi$ are Euler angles, we conclude that each rotation generates a unitary transformation with determinant 1: that is, the irreducible representation has the form of inclusion $R : \mathrm{SO}(3) \hookrightarrow \mathrm{SU}(5)$, and we denote the image of the rotation group with respect to this inclusion as $\mathfrak{G}_3$. Hence, $\mathfrak{G}_3 \subset \mathrm{SU}(5)$.

### 3. SO(3)-Irreducible Geometric Structure on a Five-Dimensional Hermitian Manifold

The purpose of this section is to study the properties of the quadratic form

$$K(z,z) = (z^1)^2 + (z^2)^2 + (z^3)^2 + 2\,qz^4\,z^5, \tag{38}$$

which is invariant under the irreducible representation $R$ of the rotation group, where $R : \mathrm{SO}(3) \hookrightarrow \mathrm{SU}(5)$. In the previous section, we denoted the image of this inclusion by $\mathfrak{G}_3$, and according to Formulas (36) and (37), any element of the group $\mathfrak{G}_3$ can be written as a product $R_1(t)\, R_2(s)\, R_1(v)$, where $t, s, v$ are real parameters. Hence, $\mathfrak{G}_3$ is a stabilizer of the quadratic form $K(z,z)$ in $\mathrm{SU}(5)$.

Assume that $z^A = U_B^A\, \tilde{z}^B$, where $U = (U_B^A)$ is a regular complex $5 \times 5$-matrix, is a linear transformation in the five-dimensional complex space $\mathbb{C}^5$. Then, the matrix of the quadratic form $K(z,z)$

$$K = \begin{pmatrix} 1 & 0 & 0 & 0 & 0 \\ 0 & 1 & 0 & 0 & 0 \\ 0 & 0 & 1 & 0 & 0 \\ 0 & 0 & 0 & 0 & q \\ 0 & 0 & 0 & q & 0 \end{pmatrix}, \tag{39}$$

transforms under this transformation as follows:

$$\tilde{K}_{AB} = U_A^C\, U_B^D\, K_{CD}, \quad K = \tilde{K}_{AB}\tilde{z}^A\tilde{z}^B, \tag{40}$$

or in the matrix form:

$$\tilde{K} = U^{\mathrm{T}} K U, \tag{41}$$

where $K = (K_{AB})$, $\tilde{K} = (\tilde{K}_{AB})$ are matrices of the form $K(z,z)$ in different bases of the Hermitian space $\mathbb{C}^5$, and $U^{\mathrm{T}}$ is the transposed matrix of $U$. The set of all matrices $\tilde{K}$ obtained with the help of (41) will be referred to as an orbit of the quadratic form $K(z,z)$ with an indication of a group of transformations. For example, the set of all matrices $\tilde{K}$ obtained by means of unitary transformations will be referred to as an $\mathrm{U}(5)$-orbit of $K(z,z)$. Obviously, we can consider the matrix $K = (K_{AB})$ as a second-order covariant tensor in a five-dimensional vector space, and in this case, we will talk about the $\mathrm{U}(5)$-orbit of the tensor $K = (K_{AB})$. Our aim in this section is to find properties of the quadratic form $K(z,z)$ (or of the corresponding tensor $K = (K_{AB})$) such that they will uniquely determine the orbit of this quadratic form.

First of all, it is easy to see that the tensor $K$ is symmetric and unitary and that these properties are invariant with respect to the group of unitary transformations $\mathrm{U}(5)$. Indeed for any $U \in \mathrm{U}(5)$, we have

$$\tilde{K}^{\mathrm{T}} = (U^{\mathrm{T}} K U)^{\mathrm{T}} = U^{\mathrm{T}} K^{\mathrm{T}} U = U^{\mathrm{T}} K U = \tilde{K},$$

and

$$\tilde{K}\,\overline{\tilde{K}} = U^{\mathrm{T}}\,K\,(U\,\overline{U}^{\mathrm{T}})\,\overline{K}\,\overline{U} = U^{\mathrm{T}}\,(K\,\overline{K})\,\overline{U} = U^{\mathrm{T}}\,\overline{U} = E,$$

where $\overline{U}$ is the complex conjugate matrix of $U$, and $E$ is the unit matrix. Hence, the U(5)-orbit of the tensor $K = (K_{AB})$ is an orbit of a symmetric and unitary tensor.

We recall that the determinant of the matrix of a quadratic form is referred to as a discriminant of a quadratic form. It is easy to find that the discriminant of the quadratic form $K(z, z)$ is $\epsilon$, where $\epsilon = \exp(i\,\pi/3)$ is the primitive sixth-order root of unity. But the discriminant of the quadratic form $K(z, z)$ is invariant with respect to the action of the group U(5). Indeed, we have

$$\det \tilde{K} = \det (U^{\mathrm{T}}\,K\,U) = (\det U)^2 \det K = \epsilon.$$

Hence, the U(5)-orbit of the second-order covariant tensor $K = (K_{AB})$ is an orbit of the tensor with a determinant equal to $\epsilon$.

The U(5)-invariant properties of the tensor $K = (K_{AB})$ found above do not yet uniquely determine the U(5)-orbit of this tensor in the space of U(5)-orbits of all second-order covariant tensors. In order to find additional invariant conditions, we use the following fact from the matrix calculus. It is known [22] that a symmetric and unitary complex matrix $X$, that is,

$$X = X^{\mathrm{T}} = \overline{X}^{-1},$$

can be written in the exponential form $X = e^{iY}$, where $Y$ is the real symmetric matrix. The tensor $K = (K_{AB})$ is symmetric, unitary, and $\det K = \epsilon$, and these properties are U(5)-invariant. Thus, in any orthonormal basis for the five-dimensional complex space $\mathbb{C}^5$, or in other words, at any point of U(5)-orbit, this tensor considered as a matrix can be written in the exponential form $K = \exp(iS)$, where $S$ is a real symmetric matrix. Particularly in the case of matrix (39), a straightforward computation gives the block form of the fifth-order real symmetric matrix $S$:

$$S = \left(\begin{array}{c|c} 0 & 0 \\ \hline 0 & \Sigma \end{array}\right), \quad \Sigma = \left(\begin{array}{cc} \frac{\pi}{6} & \frac{\pi}{2} \\ \frac{\pi}{2} & \frac{\pi}{6} \end{array}\right). \tag{42}$$

In the particular case for which $U = (U_B^A)$ is a real unitary transformation—that is, $U^{\mathrm{T}} = \overline{U}^{-1} = U^{-1}$—we can easily find a transformation law of the matrix $S$. Indeed, in the case of a real unitary matrix $U$, we have

$$\tilde{K} = U^{\mathrm{T}}\,K\,U = U^{-1}\,K\,U = U^{-1}\,e^{iS}\,U = e^{i\,(U^{-1}\,S\,U)},$$

and $\tilde{K} = \exp(i\tilde{S})$ implies $\tilde{S} = U^{-1}\,S\,U$. But a real unitary matrix is an orthogonal real matrix, and making use of a transformation $\tilde{S} = U^{-1}\,S\,U$, the real symmetric matrix $S$ can be put into a diagonal form. Straightforward computation gives the diagonal forms of matrices $\tilde{S}, \tilde{K}$:

$$\tilde{S} = \left(\begin{array}{c|c} 0 & 0 \\ \hline 0 & \tilde{\Sigma} \end{array}\right), \quad \tilde{\Sigma} = \left(\begin{array}{cc} -\frac{\pi}{3} & 0 \\ 0 & \frac{2\pi}{3} \end{array}\right), \quad \tilde{K} = e^{i\tilde{S}} = \left(\begin{array}{c|c} E & 0 \\ \hline 0 & \Xi \end{array}\right), \quad \Xi = \left(\begin{array}{cc} \epsilon^5 & 0 \\ 0 & \epsilon^2 \end{array}\right), \tag{43}$$

where $\epsilon^5 = -q, \epsilon^2 = q$, and $E$ is the third-order unit matrix. It is easy to verify that the sixth power of the matrix $K$ of the quadratic form $K(z, z)$ is equal to the identity matrix—that is, $K^6 = E$—and this relation is invariant under real unitary transformations of the five-dimensional complex space. It is well known that if the $n$th power of a matrix is equal to the identity matrix, then the eigenvalues of such a matrix are the $n$th roots of unity. Thus, the diagonal form $\tilde{K}$ of the matrix $K$ in (43) with the sixth-order roots of unity on the main diagonal is a consequence of the fact that $K$ or $\tilde{K}$ to the sixth power is equal to the identity matrix. We proved the following statement:

**Proposition 1.** *For any orthonormal basis $\{E_A\}$, where $A = 1, 2, \ldots, 5$, for the five-dimensional complex Hermitian space $\mathbb{C}^5$, the second-order covariant tensor $K_{AB} = K(E_A, E_B)$ determined by the quadratic form $K(z, z) = (z^1)^2 + (z^2)^2 + (z^3)^2 + 2q\,z^4\,z^5$ has the following $U(5)$-invariant properties:*

- $K_{AB} = K_{BA}$ *(symmetric);*
- $K_{AB}\,\overline{K}_{CB} = \delta_{AC}$ *(unitary);*
- $\det(K_{AB}) = \epsilon$, *where $\epsilon = e^{i\pi/3}$ is the sixth-order root of unity.*

*It also has the following properties, which are invariant with respect to real unitary transformations:*

- $K^6 = E$, *where the tensor $K = (K_{AB})$ is considered as a matrix;*
- *the eigenvalues of $K = (K_{AB})$ are $1, 1, 1, q, -q$, where $q = e^{2\pi i/3}$ is the cubic root of unity.*

This statement provides a basis for studying five-dimensional complex manifolds with a structure determined by the tensor $K = (K_{AB})$. Let $(M, h)$ be a five-dimensional Hermitian manifold, where $h$ is a Hermitian metric. A Hermitian metric $h$ makes it possible to reduce the group of non-degenerate linear transformations of a tangent space $T_x M, x \in M$ of a manifold $M$ to the group of unitary transformations $U(5)$. In other words, we can consider the principal bundle of orthonormal frames over a manifold $M$ with the structure group $U(5)$. Thus, by a tensor field on a manifold $M$, we mean a tensor defined at each point of a manifold $M$ and transformed under the action of the structure group $U(5)$. If $O_{\mathbb{R}}(5) \subset U(5)$ is the subgroup of real unitary matrices, then we can consider the sub-orbit of a $U(5)$-tensor field: that is, the tensor field transforming according to the action of the subgroup $O_{\mathbb{R}}(5)$ and this sub-orbit will be referred to as a $O_{\mathbb{R}}(5)$-tensor.

**Definition 1.** *An $SO(3)$-irreducible geometric structure on a five-dimensional complex Hermitian manifold $(M, h)$ is a second-order covariant symmetric unitary tensor field $K_{AB}$ for which the determinant is equal to the primitive sixth-order root of unity $\epsilon = e^{i\pi/3}$. Moreover, the tensor field $K_{AB}$, considered as an $O_{\mathbb{R}}(5)$-tensor field, has the eigenvalues $1, q, -q$, for which the multiplicity of the eigenvalue 1 is 3, and q is the primitive cubic root of unity $q = e^{2i\pi/3}$.*

From this definition, it follows that an $SO(3)$-irreducible geometric structure on a five-dimensional Hermitian manifold $M$ can be considered as a triple $(M, h, K)$, where $h$ is a Hermitian metric of $M$, and $K$ is a second-order covariant tensor field defined on $M$ or the corresponding quadratic form. Two triples, $(M, h, K)$ and $(\tilde{M}, \tilde{h}, \tilde{K})$, will be referred to as equivalent $SO(3)$-irreducible geometric structures on Hermitian manifolds $M, \tilde{M}$, respectively, if there exists a diffeomorphism $\psi : M \to \tilde{M}$ such that

$$h(v, w) = \tilde{h}(\psi_*(v), \psi_*(w)), \quad K(v, w) = \tilde{K}(\psi_*(v), \psi_*(w)),$$

where $v, w$ are tangent vectors to a manifold $M$, $\psi_*$ is the differential of a diffeomorphism $\psi$, and $K, \tilde{K}$ are quadratic forms induced by the tensors $K_{AB}, \tilde{K}_{AB}$, respectively.

Let us study a local structure of a manifold $M$. It follows from Proposition 1 and Definition 1 that, locally, we can choose a frame $\{E_A\}$ of vector fields $E_A$ and its dual coframe $\{\theta^A\}$ of complex-valued one-forms, i.e., $\theta^A(E_B) = \delta^A_B$, such that

- $\{E_A\}$ is an orthonormal frame: that is, $h(E_A, E_B) = \delta_{AB}$ and

$$h = \theta^1\,\bar{\theta}^1 + \theta^2\,\bar{\theta}^2 + \theta^3\,\bar{\theta}^3 + \theta^4\,\bar{\theta}^4 + \theta^5\,\bar{\theta}^5;$$

- The components of the tensor $K_{AB}$ form the following matrix:

$$K = \begin{pmatrix} 1 & 0 & 0 & 0 & 0 \\ 0 & 1 & 0 & 0 & 0 \\ 0 & 0 & 1 & 0 & 0 \\ 0 & 0 & 0 & 0 & q \\ 0 & 0 & 0 & q & 0 \end{pmatrix},$$

and the quadratic form induced by these components is

$$K = (\theta^1)^2 + (\theta^2)^2 + (\theta^3)^2 + 2q\,\theta^4\,\theta^5. \tag{44}$$

It is clear that the subgroup $\mathfrak{G}_3 \subset U(5)$ (isomorphic to the rotation group) studied at the end of the previous section is the stabilizer of the quadratic form (44). Hence, we can reduce the gauge group $U(5)$ to this subgroup and consider a $\mathfrak{g}_3$-connection one-form $\omega$ on a manifold $M$, where $\mathfrak{g}_3$ is the Lie algebra of $\mathfrak{G}_3$. We can write this $\mathfrak{g}_3$-valued connection one-form as follows:

$$\omega = \begin{pmatrix} 0 & \omega^3 & -\omega^2 & -\sqrt{2}\,\omega^1 & -\sqrt{2}\,q\,\omega^1 \\ -\omega^3 & 0 & \omega^1 & -\sqrt{2}\,\bar{q}\,\omega^2 & -\sqrt{2}\,\bar{q}\,\omega^2 \\ \omega^2 & -\omega^1 & 0 & -\sqrt{2}\,q\,\omega^3 & -\sqrt{2}\,\omega^3 \\ \sqrt{2}\,\omega^1 & \sqrt{2}\,q\,\omega^2 & \sqrt{2}\,\bar{q}\,\omega^3 & 0 & 0 \\ \sqrt{2}\,\bar{q}\,\omega^1 & \sqrt{2}\,q\,\omega^2 & \sqrt{2}\,\omega^3 & 0 & 0 \end{pmatrix},$$

where $\omega^1, \omega^2, \omega^3$ are real-valued one-forms. It is easy to see that a connection one-form $\omega$ is a skew-Hermitian: that is, $\overline{\omega}^{\mathsf{T}} = -\omega$. Then, the torsion two-form $\mathsf{T}^A$ and the curvature two-form $\mathsf{R}^A_B$ of a connection $\omega$ can be expressed as follows:

$$\mathsf{T}^A = d\theta^A + \omega^A_B \wedge \theta^B, \quad \mathsf{R}^A_B = d\omega^A_B + \omega^A_C \wedge \omega^C_B.$$

Straightforward calculation gives for the torsion

$$\begin{aligned} \mathsf{T}^1 &= d\theta^1 + \omega^3 \wedge \theta^2 - \omega^2 \wedge \theta^3 - \sqrt{2}\,\omega^1 \wedge (\theta^4 + q\,\theta^5), \\ \mathsf{T}^2 &= d\theta^2 + \omega^1 \wedge \theta^3 - \omega^3 \wedge \theta^1 - \sqrt{2}\bar{q}\,\omega^2(\theta^4 + \theta^5), \\ \mathsf{T}^3 &= d\theta^3 + \omega^2 \wedge \theta^1 - \omega^1 \wedge \theta^2 - \sqrt{2}\,\omega^3 \wedge (q\,\theta^4 + \theta^5), \\ \mathsf{T}^4 &= d\theta^4 + \sqrt{2}(\omega^1 \wedge \theta^1 + q\,\omega^2 \wedge \theta^2 + \bar{q}\,\omega^3 \wedge \theta^3), \\ \mathsf{T}^5 &= d\theta^5 + \sqrt{2}(\bar{q}\,\omega^1 \wedge \theta^1 + q\,\omega^2 \wedge \theta^2 + \omega^3 \wedge \theta^3), \end{aligned}$$

and for the curvature

$$\begin{aligned} \mathsf{R}^1_2 = \zeta^{312}, \quad \mathsf{R}^1_3 &= -\zeta^{231}, \quad \mathsf{R}^1_4 = -\sqrt{2}\,\zeta^{123}, \quad \mathsf{R}^1_5 = -\sqrt{2}\,q\,\zeta^{123}, \\ \mathsf{R}^2_3 &= \zeta^{123}, \quad \mathsf{R}^2_4 = -\sqrt{2}\,\bar{q}\,\zeta^{231}, \quad \mathsf{R}^2_5 = -\sqrt{2}\,\bar{q}\,\zeta^{231}, \\ \mathsf{R}^3_4 &= -\sqrt{2}\,\zeta^{312}, \quad \mathsf{R}^3_5 = -\sqrt{2}\,\zeta^{312}, \\ \mathsf{R}^4_5 &= 0, \end{aligned}$$

where $\zeta^{ijk}$ is a two-form defined by $\zeta^{ijk} = d\omega^i + \omega^j \wedge \omega^k$, where $i, j, k$ is a cyclic permutation of integers $1, 2, 3$. It can be proved that a connection $\omega$ is consistent with a Hermitian metric $h$ and that it preserves the tensor $K = (K_{AB})$; that is,

$$\overset{\omega}{\nabla} h = 0, \quad \overset{w}{\nabla} K = 0,$$

where $\overset{\omega}{\nabla}$ is the covariant derivative of tensor fields induced by a connection $\omega$.

## 4. Discussion

In this paper, we propose a ternary analog of the concept of skew symmetry based on the group of cyclic permutations $\mathbb{Z}_3$ and its faithful representation by the cubic roots of unity $1, q, \bar{q}$. We call this ternary analog of skew-symmetry a $\mathbb{Z}_3$ skew symmetry.

The notion of a $\mathbb{Z}_3$-skew-symmetric trilinear form proposed in the present paper is motivated by the ternary generalization of the Pauli exclusion principle proposed by R. Kerner. We remind that a trilinear form is called $\mathbb{Z}_3$-skew-symmetric if it satisfies the cyclic relation (6) and the traces zero relations (7). The cyclic relation (6) implies $\tau(x, x, x) = 0$

(but not necessarily $\tau(x, x, y) = 0$), and this can be interpreted within the framework of the ternary generalization of the Pauli exclusion principle: as the impossibility of the coexistence of three quarks with identical quantum characteristics but the possibility of the coexistence of two quarks with equal isospin values. We think that the traces zero condition (7) can also be interpreted within the quark model. As is known, the singlet $R\bar{R} + G\bar{G} + B\bar{B}$, where $R, G, B$ stand for colors of quarks, is colorless, which means it does not interact with quarks, and such a superposition of states must be zero: that is, such a gluon does not exist.

The algebraic aspect of the ternary generalization of skew symmetry was studied in a number of scientific papers [6–10,12,14], wherein this generalization was used to construct ternary algebras with generators. Then, these algebras were used to construct a generalization of the Dirac operator [3,4,12,15,16]. In this article, we study a ternary generalization of the notion of skew symmetry from the point of view of geometric structures. We consider the space of third-order covariant tensors, which are $\mathbb{Z}_3$-skew-symmetric: that is, the sum of the tensor components obtained by cyclic permutations of subscripts is equal to zero. Moreover, the trace of a tensor over any pair of subscripts must be equal to zero. We think that this requirement is a ternary analogue of the fact that in the case of a second-order skew symmetric (in the classical sense) covariant tensor, all diagonal elements (with equal subscripts) are equal to zero (and hence, the sum—that is, the trace—will be also equal to zero).

Tensors of the third-order with the properties described above are known in the representation theory of the rotation group. They form a complex ten-dimensional space, and in this space, there is a twofold irreducible representation of the rotation group. In order to split this twofold representation into two irreducible representations, we decompose this ten-dimensional space into a direct sum of two five-dimensional spaces with the help of the primitive cubic roots of unity $q = \exp(2\pi i/3), \bar{q} = \exp(4\pi i/3)$. This decomposition can be considered as some kind of duality, which is possibly related to the duality quark–antiquark. We construct the five-dimensional complex Hermitian space for which the points are identified with $\mathbb{Z}_3$-skew-symmetric covariant third-order tensors. If we consider a third-order tensor as a three-dimensional matrix, then we have a five-dimensional complex space the points of which can be identified with three-dimensional matrices. Figuratively speaking, we have a five-dimensional complex space, the points of which are three-dimensional lattices, and the components of third-order ternary skew-symmetric covariant tensors are located at the nodes of these lattices. It is possible that a geometry of this five-dimensional complex Hermitian space is an appropriate geometric model for a space of our Universe at the Planck scale.

**Author Contributions:** Conceptualization, V.A. and O.L.; methodology, V.A.; software and computation, O.L.; writing—original draft preparation, V.A.; writing—review and editing, O.L. All authors have read and agreed to the published version of the manuscript.

**Funding:** This research received no external funding.

**Data Availability Statement:** No new data were created or analyzed in this study. Data sharing is not applicable to this article.

**Acknowledgments:** The authors would like to thank Stefan Groote for his valuable comments on the text of this article and the discussion of the physical structures considered in this article. The authors also thank the reviewers for their useful comments that improved the text of the article.

**Conflicts of Interest:** The authors declare no conflicts of interest.

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
