# Peer review of "SO(3)-Irreducible Geometry in Complex Dimension Five and Ternary Generalization of Pauli Exclusion Principle"

_universe, doi:10.3390/universe10010002_

Round 1

Reviewer 1 Report

Comments and Suggestions for Authors

The paper treats about possible ternary generalization of Pauli exclusion principle. The starting point is a complex finite dimensional vector space equipped with tri-linear multiplication represented as a fourth-rank tensor with three covariant indices. Only these three covariant indices are needed to describe the property. Then the discussion of possible ternary generalisation of skew symmetry is provided. For example, such tensors are ternary alternating but not necessarily binary alternating. This makes a difference in interpretation in terms of the exclusion principle and as suggested by R. Kerner might be useful for the description of quarks.

Next, a particular space of ternary skew-symmetric 3rd rank tensors over 3-dimensional spaces, and its decomposition into irreducible components of SO(3)  group is described in more detail. The paper is well written and presents interesting results, and therefore, it is suitable for publication in this special volume.

Author Response

I thank the reviewer for his positive feedback on my article and positive assessment of the results obtained.

Reviewer 2 Report

Comments and Suggestions for Authors

In this paper, the authors studied the SO(3)-irreducible geometry in complex dimension and a ternary generalisation of the Pauli Exclusion Principle.

The paper can be of interest for both physical mathematics and theoretical physics community. But there are several aspects that are not clear to me and it would be important to address them if possible. First of all, it is commonly believed that violations or generalisations of the Pauli Exclusion Principles are related to violations, deformations or generalisations of the Spin Statistics Theorem under assumptions of Lorentz invariance, micro-causality/locality in an ordinary quantum field theory. What of such ordinary assumptions are here modified? This should be made clearer in the introduction or conclusions. Moreover, it is not clear if the q parameters are here related to the so dubbed q-models, where typically creation and annihilation operators commutators or anticommutators are modified for bosons and fermions. Finally, authors mentioned possible cosmological implications. Nevertheless it is very well known that Pauli Exclusions Principle Violations are strongly constrained by several deep underground experiments such as BOREXINO and VIP collaborations. After adding these clarifications and a major revision, I can consider the paper for the publications. 

Comments on the Quality of English Language

The quality of English is acceptable. 

Author Response

First of all, we thank the reviewer for clarifications regarding possible generalizations and violations of the Pauli exclusion principle. Indeed, the Pauli exclusion principle follows from the spin statistics theorem, for the proof of which the locality of quantum field theory and Lorentz invariance are important. From an algebraic point of view, the Pauli exclusion principle leads to such an important algebraic concept as the skew symmetry of the wave function of a quantum fermion system. The ternary generalization of the Pauli exclusion principle, proposed by Richard Kerner, is based on a new type of skew symmetry called ternary skew symmetry, based on the Abelian group of cyclic permutations of three elements. Richard Kerner points out properties of the quark model that may support his proposed ternary generalization of the Pauli exclusion principle. In his articles, he points out that the principle he proposes does not contradict Lorentz symmetry, and even shows that the new type of symmetry he proposes serves as a source of Lorentz invariance. We have supplemented the list of references with those articles by Richard Kerner in which he substantiates his generalization of the Pauli exclusion principle, and we have pointed these references in the introduction in those places, where we expain a physical motivation underlying our mathematical investigations. Note that in this article we do not develop the generalization of the Pauli exclusion principle itself, but rather study the algebraic-geometric aspect of this generalization and study the corresponding geometric structures.

As for the dubbed q-models mentioned by the reviewer, as far as we know this area, in these q-models one considers deformations of the commutation relations of the creation and anihilation operators, where q is a continuous complex deformation parameter. In our article the situation is to some extent similar, and in this aspect the reviewer is right.  Our approach can be considered as a "deformation" of classical skew symmetry, based on the group Z2 , by means of a discrete parameter q, which is a primitive cube root of unity.

Reviewer 3 Report

Comments and Suggestions for Authors

REFEREE REPORT FOR

Journal: Universe
Reference: universe-2728556
Title: "SO(3)-Irreducible Geometry in Complex Dimension Five and Ternary Generalization of Pauli Exclusion Principle"
Author(s): Viktor Abramov, Olga Liivapuu
Invitation: Nov 7, 2023
Completed: Nov 8, 2023

***

The manuscript is submitted for the special issue "The Languages of Physics - A Themed Issue in Honor of Professor Richard Kerner on the Occasion of His 80th Birthday".

The study focuses on a ternary generalization of skew-symmetry and it is suitable for this special issue. The results of the paper are interesting for the community and it can be considered for publication after a revision considering the points I listed below.

[1] Some typos should be corrected such as "a SO(3)" -> "an SO(3)"; "can be found in [12–15] to In this article"; the absence of equation (50), etc.

[2] The "Discussion" section should be divided into paragraphs.

[3] The "Introduction" section contains enough material but it is not supported by references. Generally, if some claim is not developed in the text, it should be followed by proper reference(s). The authors should revise the whole text, especially the "Introduction" section in this direction.

Comments on the Quality of English Language

The typos should be corrected.

Author Response

First of all, I thank the reviewer for his valuable and fair comments.

[1] We have corrected all the typos indicated by the reviewer, as well as the typos we found. We corrected the numbering of the formulas and now the text no longer contains the number 50 without an equation.

[2]  We have divided the Discussion into paragraphs and in some places have rewritten the Discussion so that each paragraph contains material on a specific aspect of the structures discussed in the article.

[3] We have extended the list of cited papers and included in the Introduction the appropriate references in those places where we indicate structures that motivate our research, but which we do not develop in our article, for instance, a ternary generalization of Pauli exclusion principle proposed by Richard Kerner or a generalization of the Dirac operator.

Round 2

Reviewer 2 Report

Comments and Suggestions for Authors

The author replied to my comments and therefore I am ready to accept this paper for the publication.